# Linalool-based silver nanoconjugates as potential therapeutics for glioblastoma: *in silico* and *in vitro* insights

Hina Manzoor[1], Muhammad Umer Khan[1]*, Samiullah Khan[2], Mohibullah Shah[3], Chaudhry Ahmed Shabbir[4], Hamad M. Alkhtani[5]

1 Institute of Molecular Biology and Biotechnology, The University of Lahore, Lahore, Pakistan, 2 Faculty of Health and Life Sciences, INTI International University, Nilai, Negeri Sembilan, Malaysia, 3 Department of Biochemistry, Bahauddin Zakariya University, Multan, Punjab, Pakistan, 4 Faculty of Medical and Health Sciences, The University of Adelaide, South Australia, Australia, 5 Faculty of Pharmacy, King Saud University, Riyadh, Saudi Arabia

* muhammad.umer4@mlt.uol.edu.pk, umer.khan685@gmail.com

## Abstract

Glioblastoma is the most predominant type of brain tumor, and resistance to medication has hampered the effectiveness of chemotherapy for gliomas. Acyclic monoterpene alcohol, linalool, has a range of pharmacological properties. The present study aimed to evaluate the impact of linalool and its nanoformulation on glioblastoma cell proliferation. DFT and ADMET analyses were used to initially assess the physiochemical characteristics of linalool and the produced silver nanoconjugates, LN@AgNPs. STRING database and Gene Expression Profiling Interactive Analysis (GEPIA) were used to narrow the 6 genes involved in glioblastoma and underwent for molecular docking study. Using AutoDock Vina 1.5.7, ligands were docked to the interaction site of selected targets. Top scored complexes PD-L1/Ligands and PTEN/ligands were simulated using molecular dynamics. The results revealed that LN@AgNPs produced a more stable complex, because metallic bonds are more robust and durable than hydrogen bonds, which give metals their distinctive strength and stability. To confirm the cytotoxicity of the compound against GBM cell line SF-767, linalool and LN@AgNPs were evaluated by *in vitro* study to check the expression at the $IC_{50}$ concentration of top scored selected genes. The results indicated that the cytotoxic effects of linalool and LN@AgNPs were concentration dependent. In the SF-767 cancer cell line, linalool and LN@AgNPs with $IC_{50}$ (33.14 µg/mL and 22.12 µg/mL respectively) values downregulated PD-L1 expression and increased PTEN expression. In conclusion phytocompounds conjugated with AgNPs increased cytotoxicity and inhibition index in glioblastoma cells. Therefore, LN@AgNPs may be a viable option for cancer treatment.

**Data availability statement:** All relevant data are within the manuscript and its Supporting Information files.

**Funding:** The author(s) received no specific funding for this work.

**Competing interests:** The authors have declared that no competing interests exist.

## 1. Introduction

The World Health Organization classifies glioblastoma multiforme (GBM) as grade IV glioma. It accounts for 47.1% of all central nervous system malignant tumors and has a high death rate [1,2]. Surgical resection combined with chemotherapy or radiation therapy is the primary treatment option for GBM. However, within seven months of their first diagnosis, the majority of patients relapse [3]. Additionally, owing to their resistance to existing treatments, GBM patients have a substantial tumor burden. The 5-year survival rate is comparatively low, with a median survival period of 15 months, despite years of research into innovative treatments, such as immunotherapy and molecular-targeted therapy [4]. This highlights the need for the development of novel medicines. Several methods have been attempted to identify surface protein targets in this situation. Cell surface proteins are encoded by 10–20% of all human genes, and because of their subcellular location, they are great targets for cancer diagnosis and treatment [5]. Inhibitors of programmed cell death-ligand 1 (PD-L1) and programmed cell death receptor (PD-1) have demonstrated great preclinical and therapeutic promise for the treatment of cancer [6–8]. Recent developments have produced small molecule inhibitors of the PD-1/PD-L1 pathway, some of which are currently undergoing clinical trials [9]

The PI3K/AKT/mTOR pathway may participate in the regulation of PD-L1 expression. Abnormal PI3K/AKT/mTOR pathway activation results in increased PD-L1 protein translation, whereas PD-L1 overexpression can activate the PI3K/AKT/mTOR pathway inversely [10,11].

Nature possesses an abundance of active principles that have cured the human population since ancient times. Natural products comprise a wide range of chemically varied entities that derive from various sources, including bacteria, fungi, plants, and marine animals [12]. Natural products have provided the current pharmaceutical industry with the ability to create well-linked therapeutic molecules. Though they can have complicated structures in two or three dimensions while still being capable of absorption and metabolism within the human body [13]. They provide a lower risk of side effects compared to synthetic drugs, making them a safer option for long-term use. Secondly, these natural agents often target multiple pathways in the body, providing a more comprehensive therapeutic effect [14]. Plants are the source of a wide range of naturally occurring compounds known as phytochemicals. [15]. Natural compounds generated from plants are often evaluated for their potential anticancer action since they have been viewed as potential anticancer medicines. Monoterpenes are natural compounds obtained from plants that have been shown to have enormous therapeutic potential. They have been shown to have antibacterial, anticancer, and anti-inflammatory properties [16]. Linalool is a common monoterpene derived from plant essential oils. Importantly, the FDA authorized linalool and categorized it as generally regarded safe for use as an animal and human direct food additive [17,18]. The anti-inflammatory, analgesic, local anesthetic, antiviral, and antimicrobial properties of linalool have been

demonstrated [19–22]. According to studies, linalool may prevent many types of human cancer cells from growing [23,24].

Nevertheless, the naïve or free form of the linalool molecule has a number of physicochemical flaws and restrictions, including physical and chemical instability, susceptibility to degradation because it is a monoterpene, and evaporative behavior [25]. To overcome these restrictions and provide linalool with superior physicochemical properties, linalool nano-formulations have been created [26,27]. Additionally, linalool nanoformulations resulted in improved bio-distribution and increased bioavailability of linalool *in vivo* [28].

Nanoparticles smaller than 100 nm can be quickly created for targeted administration of bioactive pharmacological drugs. These platforms show an improved depth of penetration into tissues *in vivo* and increased cellular absorption *in vitro* [29]. Formulations based on nanoparticles offer a stable, affordable, and biocompatible system that is easily scalable and enhances the solubility of poorly soluble pharmaceutical molecules [30].

A crucial part has been performed by computer-aided drug discovery (CADD) and design techniques in the creation of small compounds with important medicinal benefits. CADD provides a viable way to optimize and accelerate the drug development process, with its low-cost alternatives and reduction of the dangers associated with conventional methods. These early efforts at screening can increase success rates and reduce the amount of time needed to screen candidates for medications [31,32].

This study aimed to identify the expression of upregulated and downregulated proteins in glioblastoma. Additionally, an *in silico* study was conducted to evaluate the therapeutic potential of a nanoformulation of a selected compound with the goal of narrowing down targets for subsequent *in vitro* studies.

## 2. Materials and methods

The sequential steps involved in this study are shown in Fig 1.

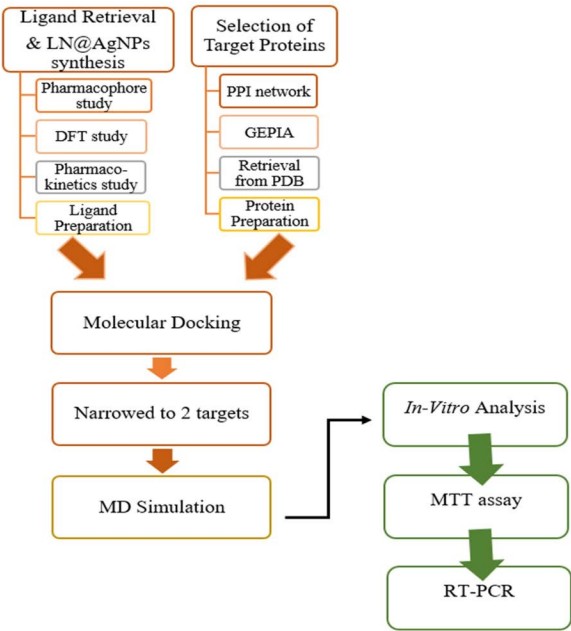

**Fig 1. Sequential experimental steps undertaken in the current study.**

## 2.1. Ethical approval

It has been confirmed that this research project was ethically approved by the institution ethical review board (Ref-IMBB/BBBC/24/904) and it also compiles the relevant institutional, national, and international guidelines and legislation with appropriate permissions from Authorities of the Institute of Molecular Biology and Biotechnology, The University of Lahore, 54000, Lahore, Pakistan.

## 2.2. Ligand retrieval and its nanoformulation

Linalool was selected to design linalool-silver nanoconjugate (LN@AgNPs) and used to assess its potential through *in silico* and *in vitro* studies.

## 2.3. *In silico* analysis

**2.3.1. Pharmacophore models for linalool.** A pharmacophore is a molecular framework that describes the essential characteristics of a molecule that give rise to its biological activity. They can be used to identify new compounds that are anticipated to be active and satisfy the pharmacophore requirements [33]. We used Biovia Discovery Studio's auto-pharmacophore generation technique to create a ligand-based pharmacophore model. The most selective pharmacophores were chosen using the Genetic Function Approximation (GFA) model, which was trained on 500 pharmacophore models. The selected models should be the most effective at differentiating between actual active hits and random matches from the MiniMaybridge database because these models are designed to optimize selectivity [34].

**2.3.2. Density functional theory (DFT) studies.** A slightly modified version of a previously reported procedure was used to perform the DFT computations [35]. All computations in the split-valence polarization (SVP) basis set were performed using the B3LYP function of the Gaussian 06 package (Rev. E.01) with default parameters. This theory makes it possible to compute the electronic structures of atoms and molecules efficiently. Optimal geometric parameters, global and local reactivity descriptors, frontier molecular orbitals (FMO), and molecular electrostatic potentials (MEP) were identified. Guass View 6 was used for the checks.

**2.3.3. Pharmacokinetic parameters.** Additionally, SwissADME (http://www.swissadme.ch) was used to predict the absorption, distribution, metabolism and excretion (ADME) properties of the newly designed LN@AgNPs along with individual linalool [36]. The pharmacokinetic features of the prepared in-house compound were estimated using the webserver named ADMETlab 2.0 (https://admetmesh.scbdd.com/) [37]. Features including absorption level, volume of distribution, metabolism of CYP binding, excretion, and AMES toxicity prediction were evaluated in these investigations.

The ProTox 3.0 (https://tox.charite.de/protox3/) server [38] was used for toxicity studies. ProTox 3.0 uses fragment propensities, chemical similarity, the most frequent characteristics, and machine learning algorithms to predict a variety of toxicity endpoints, including toxicity targets and adverse outcome (Tox21) pathways.

**2.3.4. Selection of target receptors.** A total of 50 genes directly involved immune checkpoints and related regulators were categorized into group 1 and other indirectly related genes were placed in group 2. These genes were extracted through a comprehensive literature review (S1 Table in S1 File) [2,5,39]. STRING [40], a database that offers functional connections between proteins, was used to build a Protein-Protein Interaction (PPI) network. The screening process was refined by integrating expression data from U-87MG glioma cells and head neck cancer, since glioblastoma and head and neck cancer overlap in molecular pathways. This process made use of the functional enrichment properties of the STRING network, such as disease-gene relationships (S2 Table in S1 File) and tissue-specific expression (S3 Table in S1 File). We were able to identify a subset of six genes using this method, which has given priority for further *in silico* research.

The list of six includes the upregulated genes PD-L1, CDK6, EGFR, and TP53, as well as the downregulated genes PTEN and CD8A. The Protein Data Bank (PDB) (https://www.rcsb.org/) database, which is maintained by the Research Collaboratory for Structural Bioinformatics (RCSB), contains these target proteins, namely CDK6 (PDB ID: 6OQL), PD-L1

(PDB ID: 5N2F), EGFR (PDB ID: 4HJO), TP53 (PDB ID: 7B49), PTEN (PDB ID: 1D5R), and CD8A (PDB ID: 2HP4). Because of their X-ray crystallographic structures, lower resolutions, and percentile scores in global validation measures, which show higher structural quality, PDB IDs were taken into consideration.

**2.3.5. Protein preparation.** The receptors were created by eliminating water molecules, heteroatoms, and native ligands using the BIOVIA Discovery Studio Visualizer 2021. AutoDock Tools version 1.5.7 was used to add Gesteiger partial charges and polar hydrogen atoms to the pdbqt protein file [41–43].

**2.3.6. Active site identification.** Using BIOVIA Discovery Studio 2021, the "receptor cavity method" was used to predict the binding locations of the receptor proteins. Identification and characterization of protein structure binding sites were made possible by the SDB-Site module in the BIOVIA Discovery Studio program. The inhibitory characteristics of residues found in the binding sites with center-x, y, and z values were used in this procedure [44].

**2.3.7. Ligand preparation.** The linalool structure was retrieved from the structure data file (SDF) format of the NCBI PubChem database (https://pubchem.ncbi.nlm.nih.gov/). ChemDraw Professional 16.0 was used to manually design the 2D structures of the LN@AgNPs. Chem3D 16.0 was then used to convert the 2D structures into 3D structures [45]. The files were saved as an SDF and the energy was minimized using the Chem3D Gaussian interface. To assign appropriate bond ordering, ligand preparation was performed on the SDF files of both ligands [46].

**2.3.8. Molecular docking.** AutoDock Vina 1.5.7 was used for the molecular docking [42]. After preparation, the ligand and receptor structures were moved to the Vina folder and saved in the pdbqt format. The Command Prompt (CMD) was used to launch Vina's AutoDock program [41]. "vina --config conf.txt --log log.txt" was the programming command that was run [41]. This approach determines binding affinity using a grid-based model of protein-ligand potential interactions. Soft-core potentials, which Vina tools uses, have been shown to be useful in creating a variety of random conformations of small organics and macromolecules within the active region of the target protein. To find potential therapeutic candidates, ligands were docked to the proteins and were then scored for their relative strength of interaction [43]. The co-crystallized ligand was re-docked to determine its binding affinity for the target protein before docking the test ligands [47]. The ligand interaction tool Discovery Studio 2021 was used to view the interaction diagram of the active-site residues of ligand proteins complexes [48,49].

**2.3.9. Molecular dynamic simulation.** Molecular dynamics (MD) simulations were performed on relevant ligand-protein complexes to examine the stability of ligand binding [50]. To assess the dynamic properties, the complex was loaded into the system and exposed to ff14SB conditions in AMBER. The ligand was simultaneously subjected to a generalized AMBER force field. The LEaP technique and counterion 2Cl-were used to neutralize the proteins that contained protonation. The complex was solvated by the system at an edge distance of 9.0. The resulting compound was stored in the PDB format, and the LEaP technique was used to generate coordinates and parameters. To remove the effects of stearic acid, the amount of the substance was decreased three times. The protein and ligand were then optimized by solvation and ionization during the initial minimization step. Protein and backbone amino acids were among the optimized pocket residues. The entire system was turned on to ease the complex containing the protein during the last minimization phase. After the minimization phase was completed, the system was placed within the heating panel, where the temperature was progressively increased. Following system stabilization, 100 ns of MD production utilizing the NPT ensemble at 1 atm and 300 K was performed. Ultimately, using another module, CPPTRAJ of the AMBER20 software, a number of metrics, including RMSD and RMSF, were examined following the conclusions of MD simulations with particular complexes [51]. Using the method of Arantes et al., [52] we also computed the dynamic cross-correlation matrix (DCCM) and radius of gyration (Rg) during the first 100 ns of the poses. Binding free energy (BFE) was calculated on AMBER20 in integration with MMPBSA/MMGBSA modules [53].

## 2.4. *In vitro* analysis

**2.4.1. Material.** Linalool (97% purity), Dulbecco's Modified Eagle Medium (DMEM), 3-(4,5-dimethylthiazol-2-yl)-2,5diphenyltetrazolium bromide, Dimethyl sulfoxide (DMSO), Phosphate Buffered Saline (PBS), Trypsin-EDTA, Fetal

Bovine Serum (FBS), Penicillin-Streptomycin solution were acquired from Sigma-Aldrich (Merck Group, Germany). All the reagents were analytically pure and did not require additional purification. LN@AgNPs were prepared in laboratory of University of Lahore. TRIzol reagent, PCR master mix (SYBR Green Mix) and Primers were purchased for RT-PCR analysis.

**2.4.2. Cell line.** Human glioblastoma SF-767 cell lines was obtained from cultural lab of The University of Lahore, Lahore, Pakistan.

**2.4.3. Cell culture.** The cells were maintained in Dulbecco's modified Eagle's medium at 37°C in an incubator with 5% $CO_2$. Next, 100 units/ml penicillin, 100 μg/mL streptomycin, 1 mM sodium pyruvate, 1 mM nonessential amino acids, and 10% fetal bovine serum were added [17].

**2.4.4. Treatment.** To achieve 80±5% confluence, the cells were planted (104 cells/well) on 96-well plates and incubated at 37 °C, 21% $O_2$, and 5% $CO_2$. Then, 50 μL of fresh medium per well was used instead of the liquid medium. The cytotoxic effects of LN@AgNPs and free linalool were examined at doses of 3.13, 6.25, 12.5, 25, 50, and 100 μg/mL by adding 100 μL/well of successive dilutions. Dimethyl sulfoxide (DMSO; 100 μL) was added to each well of the control group. The treated plates were incubated for twenty-four hours [54].

**2.4.5. Cell viability and cell death.** The MTT assay was used to evaluate cell viability following treatment. The plates were incubated for four hours after 50 μL/well of MTT solution (0.5 mg/mL) was added (the chamber was filled with 60% oxygen at normal pressure). After disposing the liquid media, 200 μL/well of DMSO was added to dissolve the formazan crystals. The optical density (OD) of the wells was measured at 570 nm using a plate reader, and cell viability was calculated as OD sample/OD control × 100. Three runs of the tests were conducted, and the mean and standard deviation were calculated [54].

**2.4.6. Detection of PD-L1 and PTEN expression using realtime- polymerase chain reaction (RT-PCR).** Following the manufacturer's recommendations, RNA was extracted using TriPure reagent, and a UV spectrophotometer was used to determine the concentration. For RT-PCR, five micrograms of extracted RNA was used. Mouse leukemia virus reverse transcriptase was inactivated for five minutes at 95 °C, denatured for 40 cycles at 95 °C for 30 s, annealed for 30 s at 59 °C, extended for 30 s at 72 °C, and then extended for 10 min at 72 °C. β-actin was used as an internal control. Ethidium bromide staining, software analysis, and 1.5% agarose gel electrophoresis were used to analyze the data [55]. In this experiment primers used PD-L1 (233 bp), 5'-ACCTGGCTGCACTAATTGTC-3' (forward) and 5′-AATTCGCTTGTAGTCGGCAC-3' (reverse); PTEN (539 bp), 5′-ACCTGAATACTGTCCATGTGGA-3′ (forward) and 5′-GGAGTCCAGGAAATGATATCACA-3′ (reverse).

**2.4.7. Statistical analysis.** Data were statistically analyzed (mean±SD) with a 95% confidence interval using GraphPad Prism software (GraphPad, San Diego, CA, USA) [54,55]. The $IC_{50}$ values were calculated using non-linear regression parameter [54].

## 3. Results

### 3.1. *In silico* study

**3.1.1. Pharmacophore models for linalool.** Based on the acceptor/donor hydrogen bonds and hydrophobic characteristics, the Biovia Discovery Studio Auto Pharmacophore Generation module produced two highly selective pharmacophore models for linalool (Fig 2).

The pharmacophoric properties are represented by the colored spheres in the two models. In all the models, the green sphere indicates areas of the ligand that are expected to donate hydrogen bonds, the pink sphere indicates areas that can accept hydrogen bonds, and the blue spheres indicate hydrophobic characteristics. These characteristics were found on the ligand in the same or comparable spatial locations in both pharmacophore models. This implies the existence of a dual-functional group that can interact with the target proteins by both giving and receiving hydrogen bonds. This dual

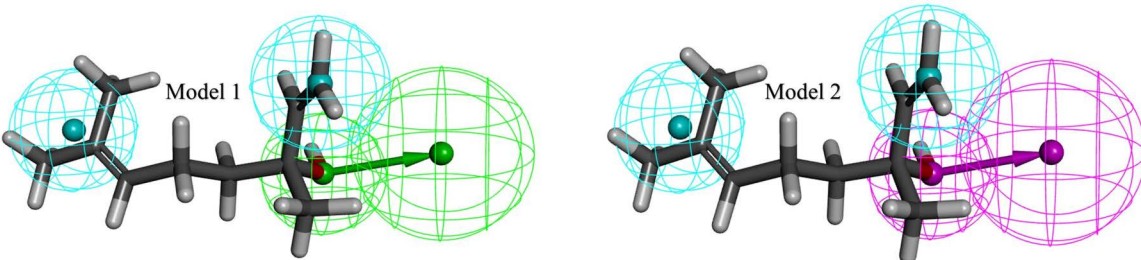

**Fig 2. Generated pharmacophore models using auto pharmacophore generation module of BIOVIA studio.**

activity may increase the ligand binding flexibility by enabling it to interact with various target sites that support the acceptance or donation of hydrogen bonds.

**3.1.2. Density functional theory (DFT) studies.** The reactivity of linalool and its synthetic silver conjugates (LN@AgNPs) was predicted using frontier molecular orbital analysis (HOMO-LUMO) (Fig 3). Additionally, the energy gap (ΔEgap) aids in characterizing kinetic molecular stability and chemical reactivity.

The dipole moment, lowest unoccupied molecular orbital (LUMO) energies, highest occupied molecular orbital (HOMO) energies, and anticipated total energies for the linalool ligand and its silver nanoconjugates are displayed in Table 1. Complexes are more stable than free ligands because the total energy of linalool-silver nanoconjugates is more negative than that of free linalool. The ability of a compound to take electrons is described by the LUMO energy, whereas the HOMO energy indicates its ability to donate electrons. The kinetic stability and chemical reactivity of the molecule were described

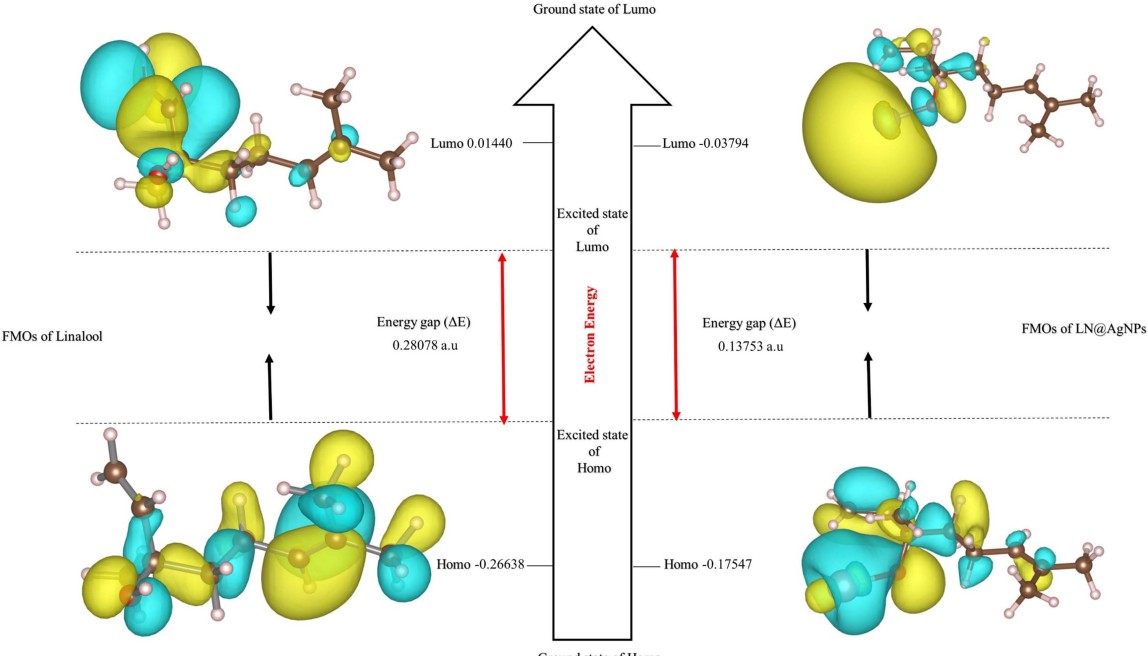

**Fig 3. Frontier molecular orbitals (FMOs) of linalool and LN@AgNPs in the form of HOMO, LUMO and the Energy gap (Eg) using the Gaussian 06 package.**

**Table 1. Calculated energies and properties of linalool and LN@AgNPs.**

| Properties | Calculated Energies | |
|---|---|---|
| | **Linalool** | **LN@AgNPs** |
| Total energy E (a.u) | −464.539 | −5640.200 |
| Homo (eV) | −7.248 | −4.775 |
| Lumo (eV) | 0.392 | −1.032 |
| Egap = Elumo − Ehomo (eV) | 7.640 | 3.742 |
| Dipole moment (Debye) | 2.106 | 6.182 |
| Ionization potential I = -Ehomo | 7.248 | 4.775 |
| Electron affinity A = -Elumo | −0.392 | 1.032 |
| Electronegativity χ= (I+A)/2 | 3.428 | 2.903 |
| Hardness η= (I−A)/2 | 3.820 | 1.871 |
| Softness S=1/2η | 0.131 | 0.267 |
| Electrical potential μ=−χ | −3.428 | −2.903 |
| Electrophilicity ω=μ²/2η | 1.538 | 2.253 |

in part by the HOMO. Because of ligand-to-metal-ion chelation, the energy gaps (Eg) = ELUMO − EHOMO were lower for LN@AgNPs than for free linalool (Table 1 and Fig 3). The charge transfer interactions during complex formation are explained by the lower Eg values of the complexes than those of the free ligands. The following parameters were computed: electrical potential (μ), softness (S), hardness (η), electrophilicity (ω), ionization energy (I), electron affinity (A), and electronegativity (χ) (Table 1).

**3.1.3. Molecular electrostatic potential (MEP) studies.** The reactive behavior of a molecule has also been described using MEP surface designs, where the positive and negative sectors are nucleophilic and electrophilic spots, respectively. Both nucleophilic cores and electrophilic sites were positive (blue) and negative (red) sectors over the molecules during the current investigation (Fig 4).

According to the analysis, both complexes may be tightly connected to the negative molecular electrostatic potential of the targets being researched, which revealed that they had a high positive electrostatic potential distributed throughout their skeleton. The development of a stabilizing complex for optimal drug docking within the ligand binding of the targets under study may benefit from this conclusion.

**3.1.4. Pharmacokinetic parameters. Analysis of physicochemical properties and drug-likeness:** SwissADME was used to examine the physiochemical and drug-likeness characteristics of linalool and its nanoconjugates. The

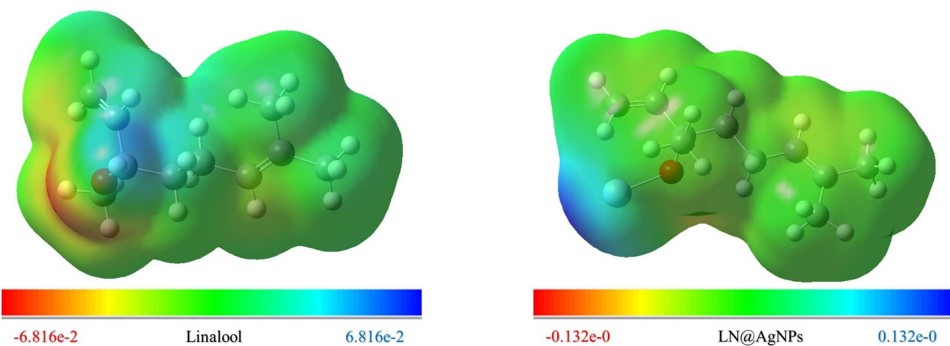

**Fig 4. The Molecular electrostatic potential surface of selected compound linalool and its nanoconjugate (LN@AgNPs).**

**Table 2. Physiochemical and drug likeness properties of selected compounds.**

| Physiochemical Properties | | | | | | | |
|---|---|---|---|---|---|---|---|
| Compounds | MW | n HBA | n HBD | n Rotb | LogP (≤5) | TPSA | MR |
| | (≤500 da) | (≤10) | (≤5) | | | < 140 Å2 | |
| Linalool | 154.25 | 1 | 1 | 4 | 3.21 | 20.23 | 50.54 |
| LN@AgNPs | 261.11 | 1 | 0 | 5 | 2.86 | 9.23 | 49.3 |
| **Drug Likeness** | | | | | | | |
| Compounds | Lipinski | Ghoose | Veber | Egan | Bioavailability Score | | Violations |
| Linalool | Yes | No | Yes | Yes | 0.55 | | 0 |
| LN@AgNPs | Yes | Yes | Yes | Yes | 0.55 | | 0 |

physiochemical characteristics were demonstrated by the data displayed in Table 2. The TPSA (total polar surface area) values for Linalool and LN@AgNPs were 20.23 Å and 9.23 Å, respectively, indicating that the linalool and its synthesized nanoconjugates fell within the Lipinski rule of five (LRF) cut-off range. There was also no violation (with the exception of Linalool's Ghoose violation) in any of the other parameters used by Ghose, Veber, and Egan, further indicating the drug likeness of both candidates.

**ADME analysis:** Fig 5, shows a BOILED-Egg and the absorption of both candidates within the body. These chemicals were not linked to increased drug resistance, as the figure demonstrates that they are not substrates of P-glycoprotein (PGP). It was also observed that because of its small size, the phytochemical linalool and its nanoconjugate has the capacity to pass through the blood-brain barrier (BBB). All the associated parameters (i.e., flexibility, lipophilicity, size, polarity, insolubility, and instauration) fell within the designated red zone, as demonstrated by the radar plots in Fig 6a,b, suggesting a suitable candidate for oral drug delivery.

The results illustrated in Table 3 showed the main ADME descriptors involved in the absorption, distribution, metabolism, and excretion of a drug.

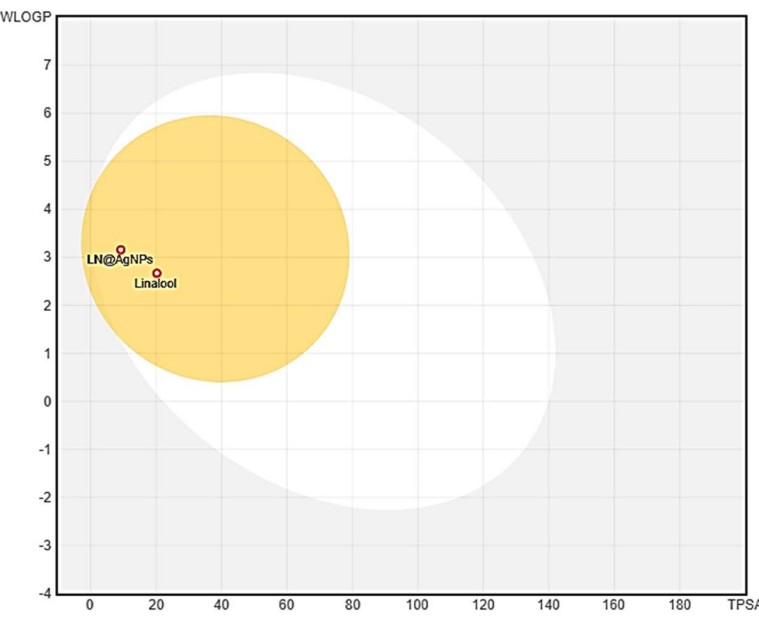

**Fig 5. BOILED-Egg results of two selected compounds in comparison, generated by SwissADME server.**

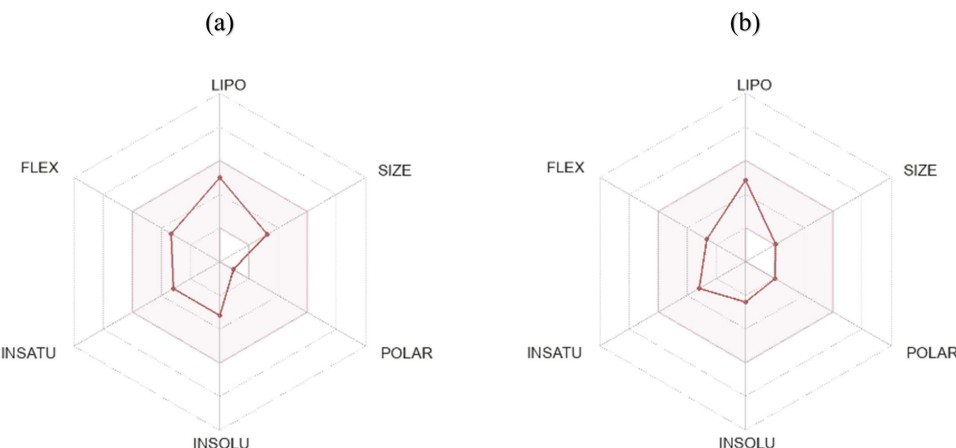

(a)                                                    (b)

**Fig 6. Bioavailability radar of selected compounds linalool (a) and LN@AgNPs (b) using the SwissADME server.**

**Table 3. The absorption, distribution, excretion, metabolism, and toxicity of linalool and LN@AgNPs.**

| Absorption | | | | |
|---|---|---|---|---|
| Compounds | Caco-2 permeability | MDCK permeability | GI absorption | Pgp-substrate | HIA |
| Linalool | −4.374 | 2.40E-05 | High | No | Yes |
| LN@AgNPs | −4.393 | 2.70E-05 | High | No | Yes |
| **Distribution** | | | | **Excretion** | |
| Compounds | PPB | VD | BBB | CL | T1/2 |
| Linalool | 85.77% | 1.721 | Yes | 7.936 | 0.609 |
| LN@AgNPs | 81.64% | 1.572 | Yes | 8.172 | 0.246 |
| **Metabolism** | | | | | |
| Compounds | CYP1A2 inhibitor | CYP2C19 inhibitor | CYP2C9 inhibitor | CYP2D6 inhibitor | CYP3A4 inhibitor |
| Linalool | No | No | No | No | No |
| LN@AgNPs | No | No | No | No | No |
| **Toxicity (Toxicophore Rules)** | | | | | |
| Compounds | Acute Toxicity | Genotoxic Carcinogenicity | Skin Sensitization | Aquatic Toxicity | Non-Biodegradable |
| Linalool | No alert | No alert | No alert | 1 alert | No alert |
| LN@AgNPs | No alert | No alert | No alert | No alert | No alert |

**Toxicology profile:** Toxicological effects must be considered when assessing the possible harm of an inhibitor to the human body. The phytochemical linalool and its nanoconjugates showed considerable non-toxic behavior against all evaluated parameters, including hepatotoxicity, toxicity endpoints, and toxicity pathways, according to the toxicity data displayed in Fig 7, which were produced using ProTox 3.0. Additionally, the ADMETlab 2.0 server was used to perform toxicophore studies of both drugs (Table 3). Linalool had one alert (Aquatic Toxicity) in the toxicological profile data listed in Table 3, indicating that the linalool-silver nanoconjugate (LN@AgNPs) is a more promising candidate for use as a therapeutic candidate with zero alertness.

**3.1.5. Selection of target receptors.** 50 gene directly and indirectly involved in immune checkpoints were extracted from literature review to construct PPI network on STRING database (Fig 8).

Using the STRING network, a list of 12 genes was condensed based on their significance in U-87MG and head and neck cancer. Based on the results of Gene Expression Profiling Interactive Analysis (GEPIA), the expression of these

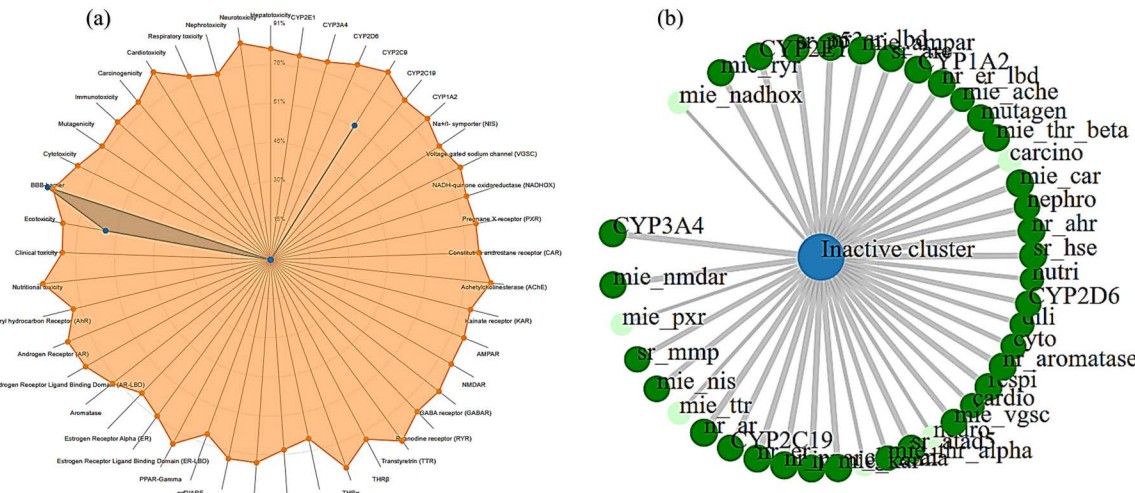

**Fig 7. The toxicity radar map shows the likelihood of positive toxicity results in relation to the average of the class (a).** The network chart shows the relationship between the chosen substance and anticipated activities (b).

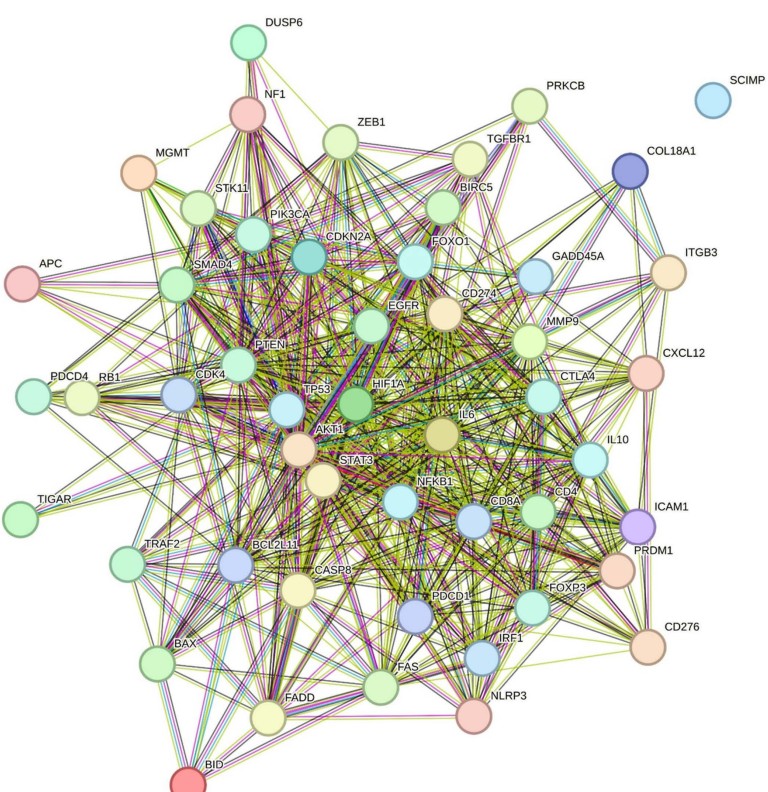

**Fig 8. PIP network of 50 gene directly and indirectly involved in immune checkpoints using STRING.**

genes in GBM and normal controls was compared using Student's t-test and displayed as scatter-box plots (Fig 9). Two somewhat downregulated genes, PTEN and CD8A, were chosen together with the four upregulated GBM genes PD-L1, CDK6, EGFR, and TP53 that had the highest Z scores (S2 Table in S1 File).

**3.1.6. Molecular docking.** First, we re-docked co-crystallized ligands (CCL) into respective proteins active site to verify the docking process. The re-docked superimposed structures of CCL of all proteins and their number of interacting residues are shown in S1 Fig in S1 File, indicating the success of docking procedures against all targets. To ascertain the binding affinities and mode of interactions of free linalool and its synthesized nanoconjugates as inhibitors, docking studies were performed on six GBM targets with the following PDB IDs: EGFR (PDB ID: 4HJO), PD-L1 (PDB ID: 5N2F), TP53 (PDB ID: 7B49), CDK6 (PDB ID: 6OQL), CD8A (PDB ID: 2HP4), and PTEN (PDB ID: 1D5R). The binding cavity of each target protein was used to dock both the ligands. The docking scores of linalool and LNAg@NPs for each target protein are shown in Table 4.

With docking scores of −5.9 kcal/mol, −6.5 kcal/mol, −4.4 kcal/mol, −5.8 kcal/mol, −4.7 kcal/mol, and −4.8 kcal/mol against all targets (4HJO, 5N2F, 7B49, 6OQL, 2HP4, and 1D5R) accordingly, the LN@AgNPs continued to be the top scorer, according to the docking experiments.

Interestingly, however, linalool and LN@AgNPs demonstrated the strongest docking score against 1D5R between downregulated and 5N2F among the upregulated genes of GBM, suggesting a superior manner of interaction compared to the other targets. The interactions between linalool and LN@AgNPs against every target, aside from 5N2F and 1D5R, were displayed using Discovery Studio Visualizer (S2 Fig in S1 File).

In the PDL1-ligands complex-bound system, graphical analysis showed that the hot residues in the ligand-binding site were Tyr56, Met115, Ile116, Ala121, Asp122, and Tyr123 (Table 5). In both the donor and acceptor motifs, linalool forms hydrogen bonds with the -NH group of isoleucine (Ile116) residue. Additionally, by using the ligand's C-H to tyrosine (Tyr56) pi-orbitals, π-sigma interactions were found to improve the stability of both complexes (Fig 10).

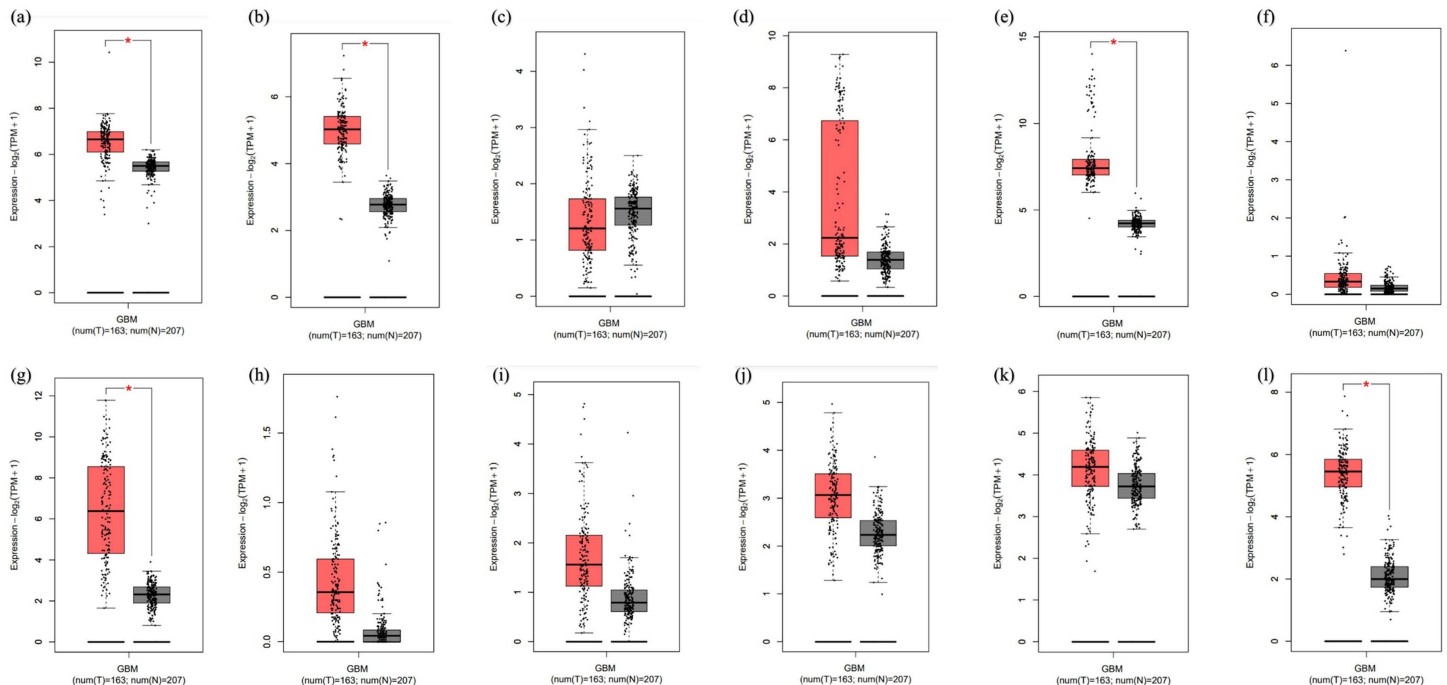

**Fig 9. Box plots showing the expression of the 12 genes identified by the Gene Expression Profiling Interactive Analysis in glioblastoma multiforme: (a) AKT1, (b) CASP3, (c) CD8A, (d) CDK2NA, (e) CDK6, (f) CLAT4, (g) EGFR, (h) PDCD1 (i) PD-L1, (j) PROM1, (k) PTEN and (l) TP53. Num, number; T, tumor; N, normal.**

**Table 4. Docking score of selected compounds against glioblastoma gene targets.**

| Compounds | Target Proteins | | | | | |
|-----------|-----------------|---|---|---|---|---|
| | 4HJO (kcal/mol) | 5N2F (kcal/mol) | 7B49 (kcal/mol) | 6OQL (kcal/mol) | 2HP4 (kcal/mol) | 1D5R (kcal/mol) |
| Linalool | −5.5 | −5.8 | −4.5 | −5.6 | −3.6 | −4.4 |
| LN@AgNPs | −5.9 | −6.5 | −4.4 | −5.8 | −4.7 | −4.8 |

**Table 5. Residual amino acid interactions (H-bond, metal acceptor and hydrophobic).**

| Receptor Protein | Ligand | Metal Acceptor | H-Bonds | H-Bonds Residues | Hydrophobic |
|------------------|--------|----------------|---------|------------------|-------------|
| 5N2F | Ln@AgNPs | Ile116, Asp122 | -- | -- | Tyr56, Ala121, Met115, Tyr123 |
| | Linalool | -- | 2 | Ile116 | Tyr56, Ala121 |
| 1D5R | Ln@AgNPs | Tyr16, Arg159 | – | -- | Ala126, Tyr16 |
| | Linalool | -- | 0 | -- | Tyr16 |

However, by establishing robust coordination bonds with the oxygen atoms of the aspartic acid (Asp122) and isoleucine (Ile116) residues, LN@AgNPs produced a more stable complex. The oxygen atoms from aspartic acid and isoleucine provided electron pairs that coordinated with Ag (Fig 10). Because metallic bonds are more robust and durable than hydrogen bonds, which give metals their distinctive strength and stability, LN@AgNPs are more stable toward protein 5N2F with metal-acceptor bonds with residues Ile116 and Asp122. Hydrogen bonds are relatively weaker and less durable, yet are essential in biological systems and intermolecular interactions.

The complex of 1D5R with LN@AgNPs was once again more stable than free linalool when we looked at ligands for the PTEN protein (Fig 11). This is because the silver atoms in LN@AgNPs formed two metal-acceptor linkages with Tyrosine (Tyr16) and Arginine (Arg159) (Table 5).

**3.1.7. Molecular dynamic simulation.** Classical MD simulations were used to assess the dynamic stability and intermolecular interactions of linalool in the complexes with 5N2F and 1D5R as a function of 100 ns. The root mean square deviation (RMSD) of linalool in combination with PTEN (PDB ID: 1D5R) and PD-L1 (PDB ID: 5N2F) is displayed in Fig 12. Over a 100 ns simulated interval, the RMSD values of the PD-L1/linalool inhibitor combination were compared. Their dynamic stability and sampling patterns were determined using ligands, pockets, and protein RMSD. Over 100 ns, RMSD was computed using the starting shapes of the molecules. The PD-L1/linalool complex's ligand, pocket, and protein atoms were all stable (Fig 12a). With an RMSD continuously below 2 Å, the ligand exhibited even greater stability than in Plot b, suggesting that it was firmly held within the binding pocket with little mobility. The RMSD variation for protein PTEN in Fig 12b ranges from approximately 2 Å to approximately 5 Å, with an initial RMSD of 2.5 Å to 3.5 Å that progressively increased at 40 ns. The ligand is stable within the binding pocket but may undergo minor perturbations, as evidenced by its steady RMSD of 1.5 Å with sporadic dips.

Furthermore, the root mean square fluctuation (RMSF) of the ligand–protein complexes was examined. Fig 13a,b display the RMSF values for linalool in complex with 5N2F and 1D5R, respectively. Most of the RMSF values were less than 3 Å, except at the binding site, indicating that both proteins were stable during the 100 ns simulation. The C-termini of the protein tails fluctuated in both complexes. However, at approximately 6 Å, a noticeable fluctuation was observed in PTEN (PDB ID: 1D5R) close to residues 240–260 (Tyr240–Lys260). At the same time, it is significant that the PD-L1 (5N2F) fluctuations that were seen close to residues 115–125 (Met115 – Arg125) lay around 5 Å. Flexibility is necessary for conformational changing the areas to interact with ligands or other molecules.

The protein folding state, modification, and overall compactness were determined using the radius of gyration (RGyr). PD-L1 (5N2F) receptor was compact and consistent, as shown in Fig 13c. A little fluctuation was seen between 0 and 40

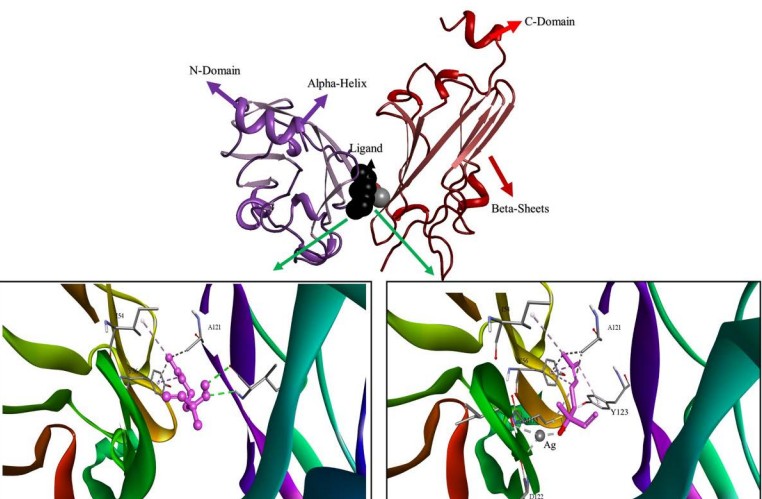

**Fig 10.  3D representation of docked complexes; binding mode of linalool and LN@AgNPs to the PD-L1.**

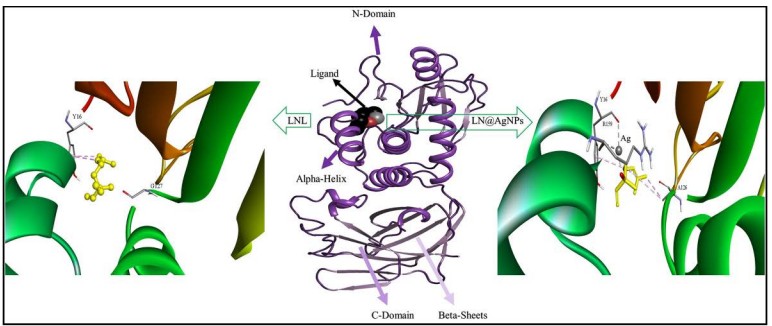

**Fig 11.  3D representation of docked complexes; binding mode of linalool and LN@AgNPs to the PTEN.**

ns, spanning from 19.8 Å to 20.5 Å. In comparison to PD-L1 (5N2F), the Rg values for PTEN (1D5R) ranged from 21.4 to 21.8 Å (Fig 13d), indicating a somewhat less compact structure.

DCCM was used to correlate the movement of the residues throughout the protein chain. For both complexes, the plots in Fig 14a,b reveal different DCCM patterns. To visualize the level of correlation between the nobilities, a color-coded scheme was developed; the blue color indicates a low correlation with the residues, whereas the red to pale green hues show highly connected mobility. Areas of high positive (red) and negative (blue) correlations were balanced in complex PD-L1/linalool's correlated motions, which may suggest flexibility and cooperative movements between certain residue pairs. Because residues have a perfect correlation with one another, self-correlation is shown by the diagonal line (red). Complex PTEN/linalool appears to exhibit a more coordinated and collective action across the protein structure. Compared to complex PD-L1, the decreased negative correlation (blue) may suggest less dynamic flexibility.

We used PCA and Gibbs free energy landscape (FEL) analyses to investigate the sub-conformational structural changes in reteplase. Fig 15 shows the PCA projections and representative conformations along PC1 and PC2 for linalool bound to PD-L1 and PTEN. Fig 16 illustrates the corresponding FEL plots, with ΔG values ranging from 0.5 to 4 kJ·mol$^{-1}$, highlighting stable and meta-stable conformational states.

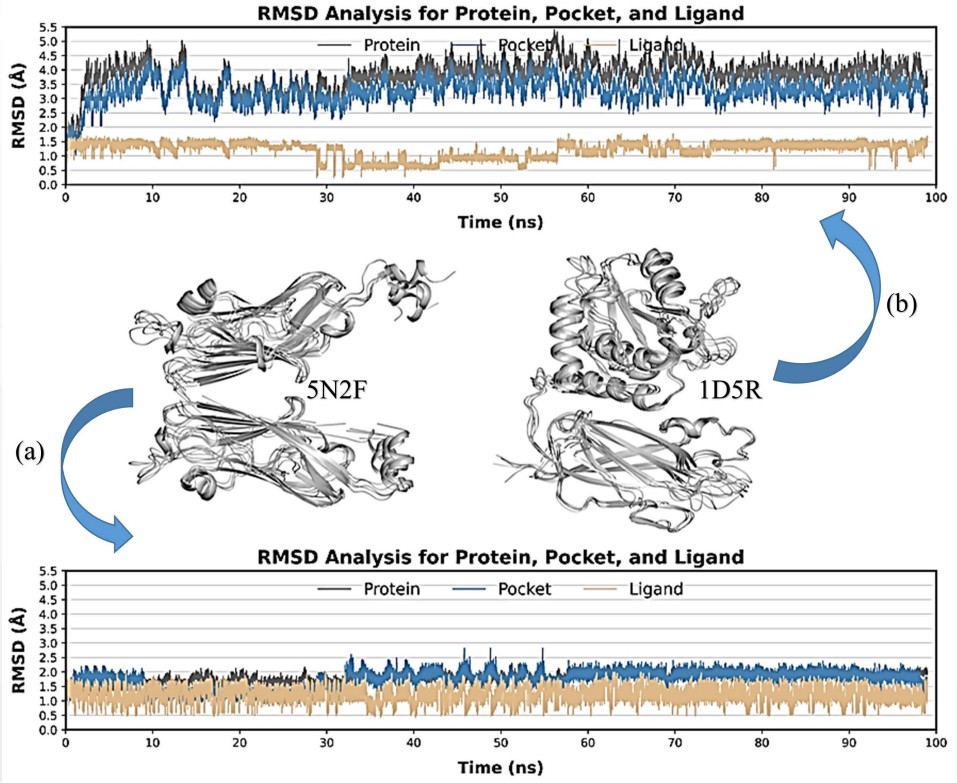

**Fig 12. RMSD plots for linalool in complex with PD-L1 (a) and PTEN (b); from each docked complex's individual 100 ns MD simulation trajectory, the ligand RMSD values were calculated as the protein-fit ligand.** Protein RMSD values were retrieved for the alpha carbon atoms.

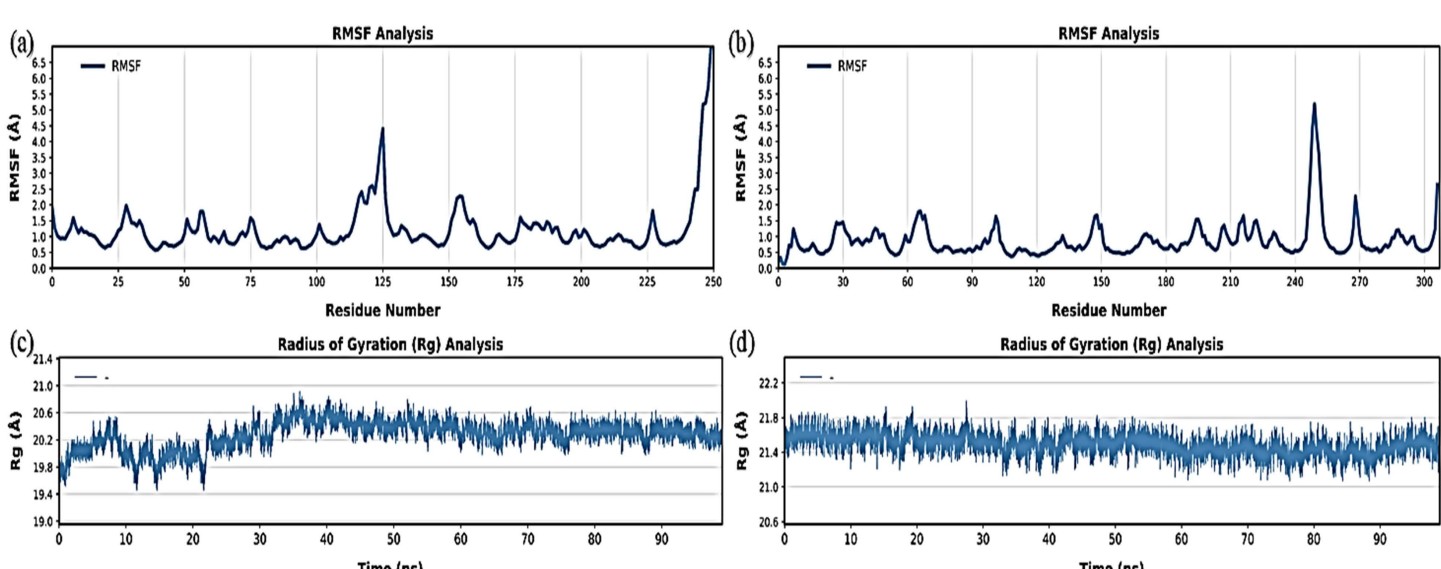

**Fig 13. Root Mean Square Fluctuation (RMSF) and Radius of Gyration (Rg) analyses of protein-ligand complexes over a 100 ns molecular dynamics (MD) simulation.** (a, b) RMSF plots showing the flexibility of each residue in the protein structures. (c, d) Rg plots showing compactness and structural stability over time. Panels (a) and (c) represent the PD-L1–linalool complex, whereas panels (b) and (d) correspond to the PTEN–linalool complex.

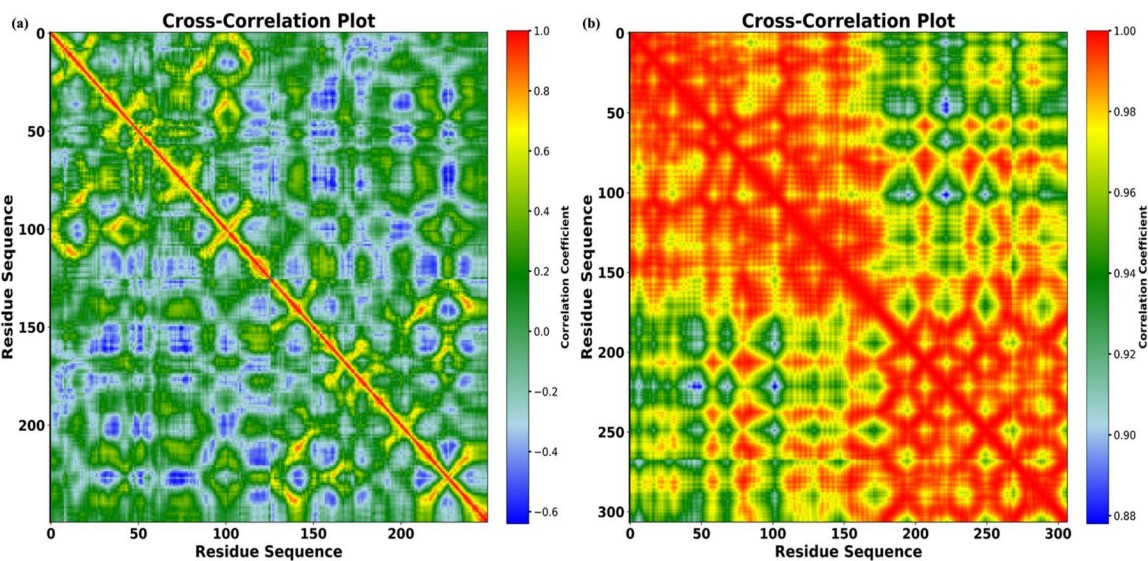

**Fig 14. Cross-correlation matrix showing coordinate fluctuations for C α atoms around the mean positions during MD simulation: positive correlations are represented by red, whereas negative correlations are represented by blue.** PD-L1/linalool (a) and PTEN/linalool (b).

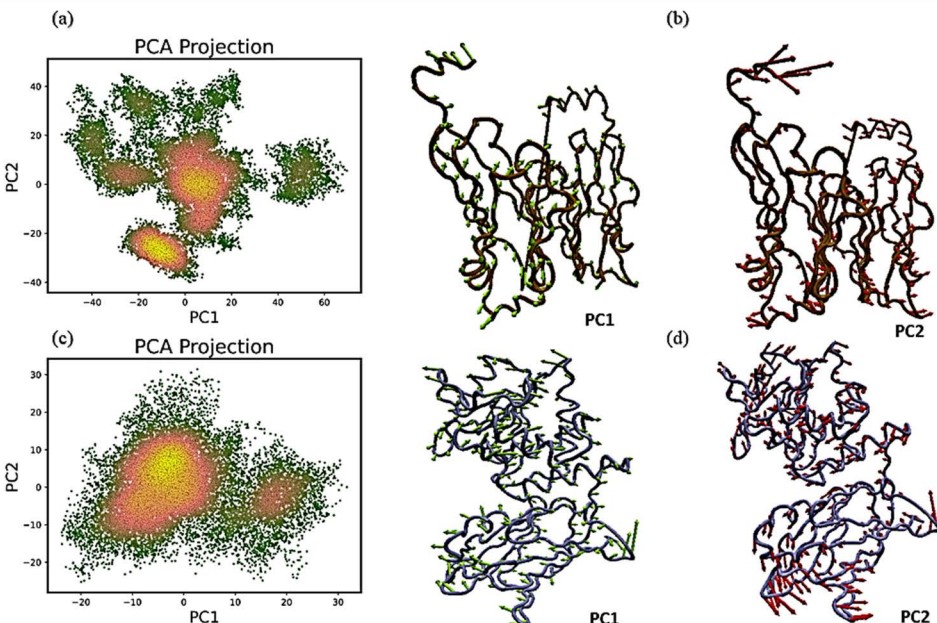

**Fig 15. Principal Component Analysis (PCA) of linalool-bound complexes.** (a, c) PCA projections showing motions along the first two principal components (PC1 and PC2) for the PD-L1 and PTEN complexes, respectively. (b, d) Representative conformations captured along PC1 and PC2, illustrating dominant motions extracted from the PCA.

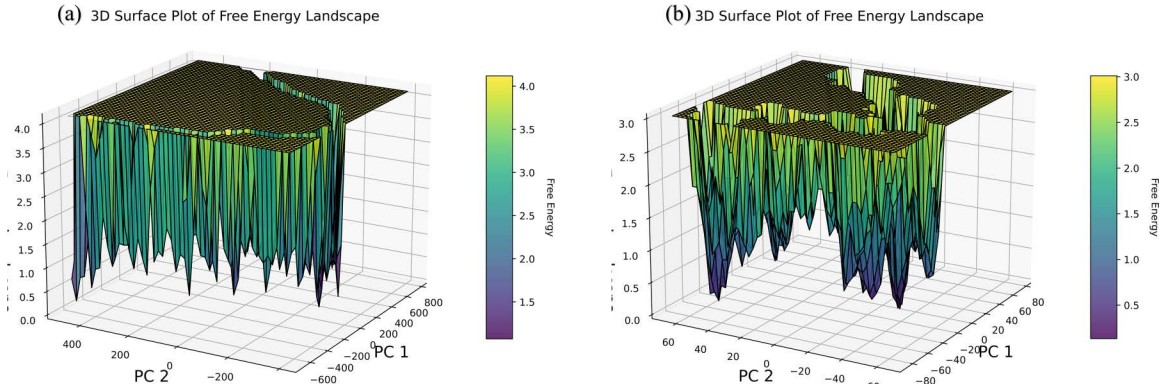

**Fig 16. Free Energy Landscape (FEL) of linalool-bound complexes.** (a, b) 3D surface plots of Gibbs free energy landscapes for PD-L1 and PTEN complexes, respectively. Purple regions represent low-energy conformations, green indicates meta-stable states, and yellow highlights high-energy conformations.

**Binding free energy calculations:** To demonstrate the stability of each system, 1000 samples were collected during the last 2 ns of the simulations, and binding free energy calculations were performed. Fig 17a,b show the binding affinities determined for both complexes using the MMGBSA and MMPBSA techniques. Despite unfavorable polar solvation, Complex PD-L1/linalool exhibits a greater binding affinity ($\Delta G_{pred}$ PB: ~−15 kcal/mol), driven by considerable van der Waals interactions ($\Delta E_{vdW}$: ~−30 kcal/mol). Complex PTEN/linalool has a lower gas-phase interaction energy and lower van der Waals contributions ($\Delta E_{vdW}$: ~−8 kcal/mol), which results in weaker binding ($\Delta G_{pred}$ PB: ~−5 kcal/mol). These results are consistent with the docking results..

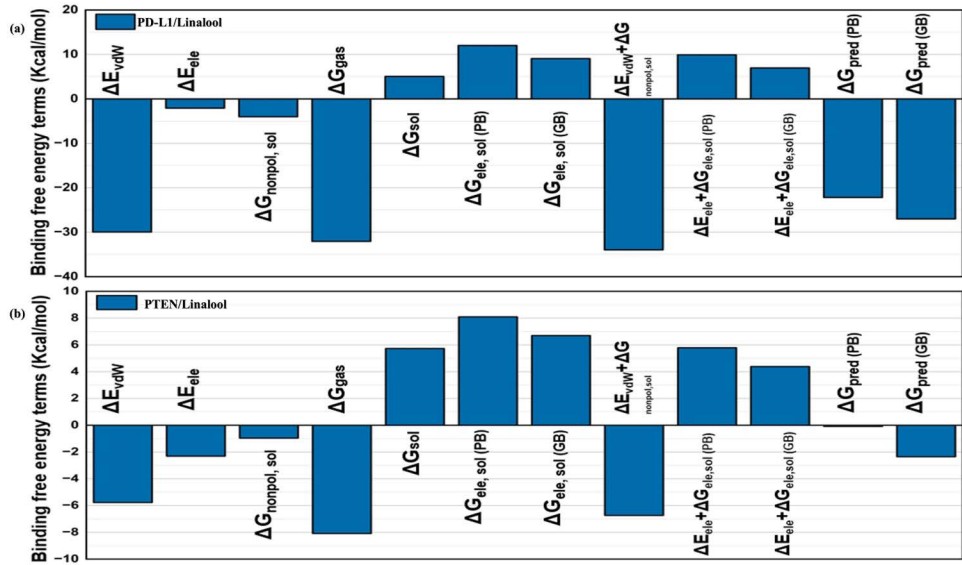

**Fig 17. Comparison for the binding free energy terms applied, PD-L1/linalool (a) and PTEN/linalool (b).**

## 3.2. *In vitro* analysis

### 3.2.1. Cell viability and cell death.

SF-767 cells were treated with 3.13–100 µg/mL of linalool and its silver-nanoconjugates LN@AgNPs for 24 h to evaluate the possible effects of these compounds on the growth of glioma cells. The effect of linalool and LN@AgNPs on cell viability and mortality was investigated (Fig 18), and statistical analysis was used to determine the $IC_{50}$ values (Fig 19). The results showed that concentrations below the therapeutic dose of the medication (15 µg/mL) did not produce noticeable cytotoxicity compared to the control group (blank). Following a 24-hour incubation period, SF-767 cells treated with LN@AgNPs ($IC_{50} = 22.12$ µg/mL) exhibited greater cytotoxicity compared to its free form, linalool ($IC_{50} = 33.14$ µg/mL). At 25, 50, and 100 µg/mL, LN@AgNPs significantly reduced viability compared to free linalool ($p < 0.0001$), suggesting a controlled and sustained anticancer effect. The nanoformulation may offer advantages over free linalool in terms of stability, targeted delivery, and prolonged therapeutic action, potentially minimizing toxicity while maintaining efficacy (Fig 18).

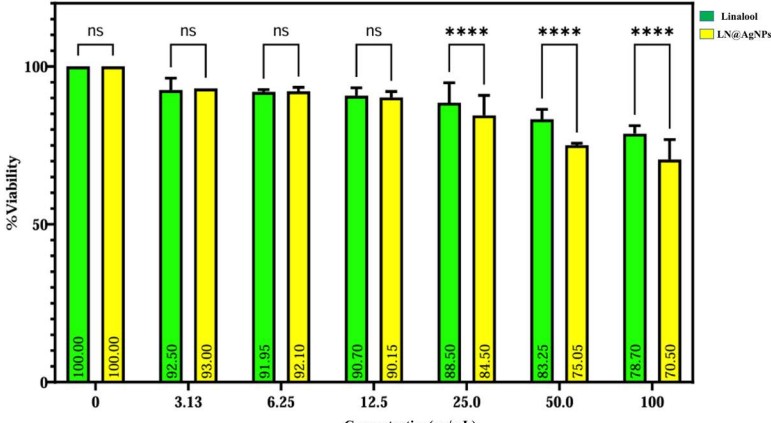

**Fig 18. SF-767 glioblastoma cell line viability after treatment lianlool (green) and LN@AgNPs (yellow) was assessed using the MTT assay and results are presented as mean ± 95% confidence interval (CI) with error bars.** Statistical significance was determined using two-way ANOVA followed by Tukey's post hoc analysis, with significance levels indicated as ns (not significant), $p < 0.05$, $p < 0.01$, $p < 0.001$, and $p < 0.0001$.

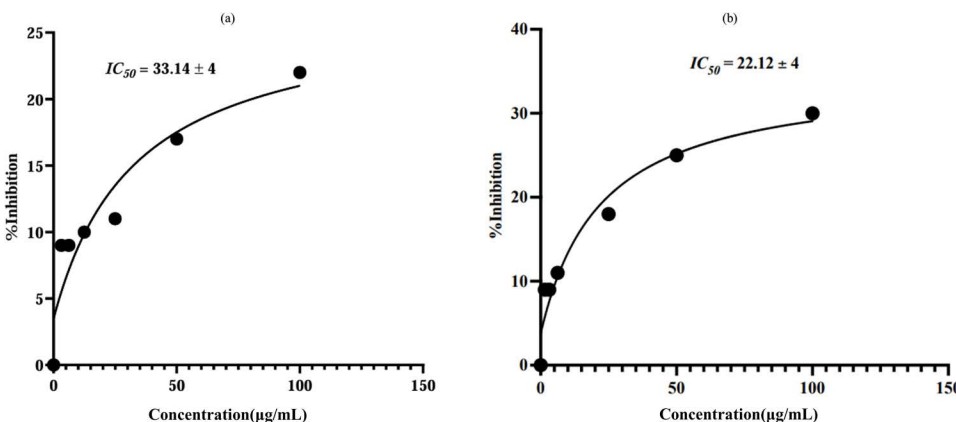

**Fig 19. Use of $IC_{50}$ values to parameterize concentration–effect curves.** A plot of the concentration–response curve for Linalool (a). A plot of the concentration–response curve for LN@AgNPs (b).

**3.2.2. Detection of PD-L1 and PTEN expression using realtime- polymerase chain reaction (RT-PCR).** After 24 h of exposure to the $IC_{50}$ concentrations of the substances, the cells were subjected to RT-PCR analysis to determine gene expression by setting the β-actin gene as a reference. In the SF-767 cancer cell line, both samples at $IC_{50}$ values demonstrated a 1.5 to 2-fold reduction in PD-L1 expression ($p \leq 0.0001$) while maintaining consistently high PTEN expression levels (Fig 20).

## 4. Discussion

Glioblastoma multiforme (GBM), a grade IV glioma according to the World Health Organization, is treated by surgical resection along with chemotherapy or radiation therapy. However, most individuals experience relapse within seven months after their initial diagnosis [3,56]. Natural compounds derived from plants are regularly being evaluated for their potential anticancer action since they have been viewed as potential anticancer medicines [57]. Linalool is frequently present in the essential oils of over 200 plant and herb species [24,57]. However, the free form of linalool has several limitations. Linalool nanoformulations have been developed to overcome these limitations and improve the physicochemical properties of compounds [26,27].

The goal of this study was to identify the target proteins implicated in the overexpression and downregulation of glioblastoma. To narrow down the target for subsequent *in vitro* evaluation of the nanoformulation, an *in silico* analysis was carried out to evaluate the potential for nanoformulation of a chosen molecule.

DFT and ADMET analyses were used to initially assess the physiochemical characteristics of linalool and the produced silver conjugates, LN@AgNPs. The reactivities of linalool and LN@AgNPs were predicted using frontier molecular orbital analysis (HOMO-LUMO) (Table 1). EHOMO and ELUMO are two examples of quantum chemical descriptors that are crucial for forecasting a molecule's energy gap, as well as its reactivity, hardness, and softness [58]. Free linalool has the highest band gap, while the computed HOMO and LUMU energy gaps for the investigated linalool molecule and its silver nanoconjugate were 7.640 eV and 3.742 eV, respectively. LN@AgNPs is a suitable contender owing to their high HOMO

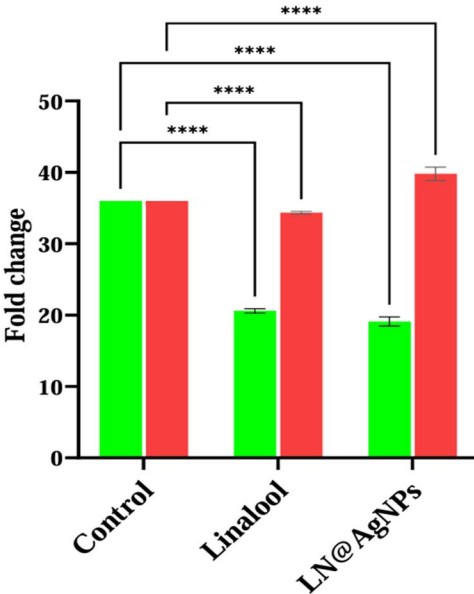

**Fig 20. Relative gene expression levels of PD-L1 (green) and PTEN (red) in cells treated with linalool and LN@AgNPs, analyzed using RT-PCR.** Data are presented as mean ± 95% confidence interval. Statistical significance was assessed using a two-way ANOVA, followed by a Tukey's post hoc test to compare each treatment group with the control.

energy (−4.775 eV), short energy gap (3.742 eV), and small chemical hardness (1.871 eV). Based on the work of Bitew et al. [58] LN@AgNPs have a high dipole moment (6.18 Debyee), which makes it a preferred ligand for biological action. The total energy of LN@AgNPs was more negative than that of the free ligands, indicating that the complexes were more stable than the free ligands, which is in contrast to the findings of El-Lateef et al. [59].

In the process of designing new drugs, pharmacokinetic analysis of compounds is essential because it forecasts how the drug will flow through the body [60]. Both substances showed excellent gastrointestinal absorption, indicating promising oral bioavailability. The pharmacological profile of potential medications is significantly influenced by P-glycoprotein (PGP) [61]. The fact that neither of these chemicals is a PGP substrate suggested that resistance will be less common in further *in vitro* investigations (**Table 3**) [62]. Prominent researchers Lipinski et al. established criteria that helped define molecular characteristics and were used to assess the drug-likeness of compounds [63–65]. Both compounds demonstrated the ability to interact with the target of interest in an efficient manner, with zero violation of these requirements (Table 2).

Furthermore, the rate at which drugs are absorbed and enter the bloodstream in their entirety is referred to as the bioavailability. Studied compounds showed good bioavailability (Fig 6a,b), these results are consistent with previous studies that stated that compounds in the pink region of the bioavailability radar have high oral bioavailability [66]. BOILED-Egg was used to evaluate the permeability of molecules via the brain barrier or up to the intestine. According to the results, substances denoted by red circles (PGP-) are good for sustaining higher concentrations in brain cells because they are not substrates of P-glycoprotein [67]. With the highest $LD_{50}$ values of 2200 and 5000 mg/kg, the candidate linalool and LN@AgNPs appeared to be hazardous only at higher doses. Increased consumption of chemicals can be harmful to the immune system [68].

The PIP network in STRING was used to compare the expression of 12 genes associated with GBM with that of normal controls based on GEPIA data using Student's t-test, displayed as scatter-box plots (Fig 9). PTEN and CD8A were downregulated in GBM tissues, whereas PD-L1, CDK6, EGFR, and TP53 genes with high string Z-scores were all upregulated in GBM (S2 Table in S1 File). GBM cells modify pathways known to be involved in pathogen defense to encourage tumor growth and elude immune surveillance. Significant upregulation of the PD-L1/PD-1 (CD274/PDCD1) [69] pathway is one of the key links between malignancies and non-cancerous cells. T-cell function is decreased, and T-cell exhaustion or death is caused by PD-L1 binding to PD-1; these interactions also promote immunological evasion [70]. Cancer is largely caused by CDKs, which are important cell cycle regulators [71]. One of the hallmarks of GBM is unchecked cell division, which may result from dysregulation of these kinases. Another important gene that is frequently dysregulated in GBM and loses its tumor-suppressive properties is TP53 [72]. Another important factor influencing GBM invasion and angiogenesis is abnormal EGFR expression [73]. One of the most commonly altered genes in human malignancies, PTEN is a tumor suppressor that regulates growth and survival [74]. Numerous functions are under control, such as energy metabolism, differentiation, survival, proliferation, and dysregulation of cell motility and structure [75]. PTEN mutation or deletion promotes cell division while reducing tumor growth and cell death [76].

Linalool and its nanoformulations were identified by Rodenak-Kladniew B [57], as therapeutic medicines for cancer therapy.

The selected targets (PDB IDs: EGFR (PDB ID: 4HJO), PD-L1 (PDB ID: 5N2F), TP53 (PDB ID: 7B49), CDK6 (PDB ID: 6OQL), CD8A (PDB ID: 2HP4), PTEN (PDB ID: 1D5R)) and core compounds (Linalool and LN@AgNPs) were used for molecular docking studies. Higher binding affinities between ligands and proteins are correlated with higher negative free-binding energies [56]. All targets had binding energies between −3.6 and −6.1 kcal/mol (Table 4). The maximum binding affinity for both free linalool and LN@AgNPs was demonstrated by PD-L1, which had docking scores of −5.8 and −6.5 kcal/mol, respectively. H-bonds (free linalool), metal acceptor bonds (LN@AgNPs), and hydrophobic interactions with PD-L1 residues Tyr 56, Met115, and Ala121 were credited. These results are in line with those of another study, in which an inhibitory binding mechanism was represented by the BMS/PD-L1 complex. Additional interactions of pi-sigma and

pi−alkyl with Met115 and Ala121, respectively, and a T-stacking contact with the distal phenyl moiety of the inhibitor were consistent with the current work. The side chain Tyr56 plugs the binding cleft on one side [77].PTEN showed the highest binding affinities for free linalool (−4.4 kcal/mol) and LN@AgNPs (−4.8 kcal/mol) among the downregulated genes. Previous findings have indicated that specific amino acids, Tyr16, Asp24, Ala47, Gly44, Val45, Asp92, His93, Ala126, Lys128, Asp162, Lys164, Asp326, and Lys330, affect the function of the PTEN protein [78]. According to the docking analysis, some of these amino acids, Tyr16, Ala126, and Asp159, were bound by the screened compounds.

MD simulations were used to further assess the binding pattern and interaction analysis of the linalool complex with PTEN and PD-L1. The RMSD plot of linalool (Fig 12) in complex with PTEN exhibited notable fluctuations, indicating a less stable interaction or protein conformation during the simulation, while linalool complex with PD-L1 showed a slight fluctuation in RMSD, ranging within 2 Å throughout the simulations, suggesting that the complex remained stable during the simulation period and may have inhibitory activity [79–81]. Important information for describing local alterations along the protein chain can be found in RMSF, Rg plots, DCCM, and PCA analysis [82]. The linalool-PD-L1 complex's greater stability raises the possibility that linalool functions as a more potent PD-L1 inhibitor than it functions as a PTEN activator. Additionally, the present analysis is consistent with a previous study that demonstrated a similar methodology for identifying drugs as PD-L1 inhibitors [83]. Later, another study showed relevant simulation results during the 150 ns trajectory [84]. Natural phytochemical inhibitors of PD-L1 were found in another experimental study [85] which resulted in the upregulation of PTEN expression.

To confirm the cytotoxicity of the compound against GBM, linalool and LN@AgNPs were evaluated *in vitro*. The cytotoxic effects of LN@AgNPs and free linalool were examined at doses of 3.13, 6.25, 12.5, 25, 50, and 100 µg/mL by adding 100 µL/well of successive dilutions. The dose range of 3.13–100 µg/mL was selected based on previously reported studies where essential oils exhibited no significant cytotoxicity up to 128 µg/mL in RAW264.7 and L929 cells, supporting the safety of using similar concentrations for biological assays [86]. Therefore, doses within this range were adopted for our cytotoxicity evaluations against SF-767 glioblastoma cells. These findings demonstrated that LN@AgNPs had significant anticancer activity against the SF-767 cancer cell line, with an $IC_{50}$ of 22 µg/mL, in contrast to $IC_{50}$ of linalool (33.14 µg/mL). By causing inhibition, free linalool and LN@AgNPs both had anticancer effects. Furthermore, the MTT assay showed that the viability of cell populations decreased as the concentration of linalool increased, indicating that the cytotoxic effects of linalool and LN@AgNPs were concentration dependent (**Fig 18**). Enzymatic breakdown, appearance of innate responses, insufficient drug concentrations in or near tumors, difficulties in detecting tumors, and obtaining regulated drug release are all significant challenges when treating brain tumors with intravenous drug delivery. In addition to producing systemic adverse effects, these issues make it challenging to prepare appropriate dosages for efficient treatment of brain tumors. However, the using nanoparticles can improve therapeutic outcomes, decrease adverse medication reactions, and increase the efficiency of drug release [87–89]. When compared cell line under investigation, AgNPs were found to exhibit very effective selective cytotoxic activity against glioma cells. Our results are in line with those of earlier research, and the generated nanomaterial has the potential to greatly enhance glioblastoma therapy. Following treatment with free linalool and LN@AgNPs, the MTT assay results demonstrated a dose-dependent decrease in the proliferation of cancer SF-767 cells, which is in line with the findings of another investigations [90].

Additionally, to confirm our molecular docking findings for PD-L1 upregulated and PTEN downregulated genes in glioblastoma, the cells were treated with $IC_{50}$ concentrations of the samples for 24 h before being subjected to RT-PCR for gene expression analysis (Fig 20). In the SF-767 cell line, both samples showed downregulated PD-L1 and elevated PTEN expression at $IC_{50}$ values. Tumor cells have been shown to produce high levels of PD-L1, downregulate costimulatory molecules and MHC, express/activate STAT3, eliminate PTEN, lower immunogenicity, and attract Tregs [91]. These outcomes align with the current study's findings. Xia et al. demonstrated that a PD-L1 inhibitor preferentially targeted miR-33a-5p, activated the PTEN signaling pathway, and inhibited the DDR process to cause radiation sensitivity in U87 MG and U251 cells. These results offer fresh perspectives on the molecular processes by which PD-L1 inhibitors enhance

radiation sensitivity in GBM [92]. According to previous studies, brain glioblastoma development and occurrence are also significantly influenced by the P13K/Akt/mTOR signaling pathway. It has been discovered that PI3K inhibitors and PD-1 blockers work in concert to treat PTEN-deficient cancers and can enhance patient outcomes. Furthermore, the PI3K-AKT-mTOR pathway can directly affect the immune response in the TME of PTEN-deficient glioblastoma [93]. Increased PD-L1 cell surface expression induced by PTEN loss results in decreased T-cell proliferation and increased apoptosis. As PTEN depletion is one of the mechanisms regulating PD-L1 expression, agents that target the PI3K pathway may improve adaptive anticancer immune responses [94]. PIK3CA-mutated PTEN-lost tumors exhibited higher levels of CD274-positivity compared to PTEN expressing malignancies or PIK3CA-wild-type PTEN-lost tumors.

Al-Nuairi et al. demonstrated that by blocking or activating proteins in several apoptotic pathways, silver nanoparticles containing *Cyperus conglomeratus* root extract caused apoptosis in cancer cell lines [90]. It is not fully known how biologically generated nanoparticles specifically destroy cancer cell lines. However, numerous investigations have connected the method of action of AgNPs with cancer cells [95]. Consistent with the results of this investigation, variations in the number of viable cells suggest that AgNP-induced anticancer activity promotes cell death [95].

This study investigated linalool-silver nanoconjugates (LN@AgNPs) targeting glioblastoma using comprehensive *in silico* and *in vitro* approaches. It uniquely combines pharmacophore modeling, molecular dynamics, and gene expression analysis (PD-L1/PTEN) to reveal enhanced cytotoxic efficacy through nanoconjugation of a phytocompound. Despite promising results, this study is limited by the absence of *in vivo* model. Future studies should focus on evaluating pharmacokinetics, biodistribution, and toxicity profiles of suitable animal models.

## Conclusion

This study evaluated the anticancer potential of LN@AgNPs and free linalool in glioblastoma cells. The PD-L1/PTEN ratio confirmed the cytotoxic effects of both linalool and LN@AgNPs on brain cancer cells. Our findings demonstrate that conjugating phytocompounds with AgNPs significantly enhances their cytotoxicity and inhibitory potential against glioblastoma. Thus, LN@AgNPs emerge as a promising therapeutic strategy for cancer treatment. This study demonstrates how natural herbal chemicals and nanotechnology, specifically AgNPs, can work in concert to increase the effectiveness of cancer treatment. However, future preclinical trials and *in vivo* investigations are necessary to confirm the safety and therapeutic effectiveness of LN@AgNPs for clinical use. There may also be synergistic advantages when investigating their combination with currently available chemotherapeutics.

## Supporting information

**S1 File. This file contains S1 Table, S2 Table, S3 Table, S1 Fig and S2 Fig.**
(DOCX)

## Acknowledgments

We are thankful to the Researchers Supporting Project number (RSPD2025R930), King Saud University, Riyadh, Saudi Arabia.

## Author contributions

**Conceptualization:** Hina Manzoor, Muhammad Umer Khan, Samiullah Khan, Mohibullah Shah, Chaudhry Ahmed Shabbir, Hamad M. Alkhtani.

**Data curation:** Hina Manzoor, Samiullah Khan, Chaudhry Ahmed Shabbir.

**Formal analysis:** Hina Manzoor, Hamad M. Alkhtani.

**Funding acquisition:** Hamad M. Alkhtani.

**Investigation:** Samiullah Khan, Mohibullah Shah.

**Methodology:** Samiullah Khan, Mohibullah Shah.

**Project administration:** Muhammad Umer Khan, Mohibullah Shah.

**Resources:** Chaudhry Ahmed Shabbir.

**Software:** Chaudhry Ahmed Shabbir.

**Supervision:** Muhammad Umer Khan.

**Validation:** Chaudhry Ahmed Shabbir, Hamad M. Alkhtani.

**Visualization:** Chaudhry Ahmed Shabbir.

**Writing – review & editing:** Muhammad Umer Khan.

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
