## [Decision Letter · Decision Letter 0]

Dear Dr. Khan,

Thank you for submitting your manuscript to PLOS ONE. After careful consideration, we feel that it has merit but does not fully meet PLOS ONE’s publication criteria as it currently stands. Therefore, we invite you to submit a revised version of the manuscript that addresses the points raised during the review process.

We look forward to receiving your revised manuscript.

Kind regards,

Mansureh Ghavam

Academic Editor

PLOS ONE

3. Please include a complete copy of PLOS’ questionnaire on inclusivity in global research in your revised manuscript. Our policy for research in this area aims to improve transparency in the reporting of research performed outside of researchers’ own country or community. The policy applies to researchers who have travelled to a different country to conduct research, research with Indigenous populations or their lands, and research on cultural artefacts. The questionnaire can also be requested at the journal’s discretion for any other submissions, even if these conditions are not met.  Please find more information on the policy and a link to download a blank copy of the questionnaire here: https://journals.plos.org/plosone/s/best-practices-in-research-reporting. Please upload a completed version of your questionnaire as Supporting Information when you resubmit your manuscript.

Additional Editor Comments (if provided):

Reviewers' comments:

Reviewer's Responses to Questions

**Comments to the Author**

1. Is the manuscript technically sound, and do the data support the conclusions?

Reviewer #1: Yes

Reviewer #2: Yes

2. Has the statistical analysis been performed appropriately and rigorously?

Reviewer #1: Yes

Reviewer #2: I Don't Know

3. Have the authors made all data underlying the findings in their manuscript fully available?

Reviewer #1: Yes

Reviewer #2: Yes

4. Is the manuscript presented in an intelligible fashion and written in standard English?

Reviewer #1: No

Reviewer #2: Yes

Reviewer #1: Linalool-Based Silver Nanoconjugates as Potential Therapeutics for Glioblastoma: In Silico and In Vitro Insights. The study is well-designed, and the experimental approach, including in silico and in vitro, is comprehensive and relevant. However, this study need a major revision.

In the introduction, authors can enrich this manuscript following this articles if relevant:

https://doi.org/10.1016/j.prenap.2025.100206;
https://doi.org/10.71193/jpci.20250005;
https://doi.org/10.1016/j.prenap.2024.100100;
https://doi.org/10.1016/j.fbio.2023.103302;
https://doi.org/10.1002/ptr.8239;
https://doi.org/10.1016/j.prenap.2024.100124;
https://doi.org/10.1002/fsn3.4318;
https://doi.org/10.1155/2024/8843855;
https://doi.org/10.1002/cbdv.202401973;
https://doi.org/10.1002/cbdv.202401904;
https://doi.org/10.1007/s12032-025-02646-z;
https://doi.org/10.23812/j.biol.regul.homeost.agents.20233711.609;
https://doi.org/10.1016/j.neulet.2024.138060;
https://doi.org/10.1016/j.sleep.2024.12.007;
https://doi.org/10.1007/s00210-024-03665-9;
https://doi.org/10.1016/j.neulet.2024.138007;
https://doi.org/10.1016/j.fbio.2024.105469;
https://doi.org/10.1016/j.ntt.2024.107403;
https://doi.org/10.1007/s00210-025-03915-4;
https://doi.org/10.1016/j.neuroscience.2025.03.004;
https://doi.org/10.1002/brb3.70446;
https://doi.org/10.1111/cns.70350;

Draw a figure for method section that’s why reader can easily understand.

In results section figures, authors should add individual data points.

Authors should add novelty, and study limitation section before conclusion.

Authors should be add a possible mechanism for this study.

Authors should be add a section of docking validation.

Mention Chemicals and reagents names.

Mention the concentration selection process.

Write the table and figure legends.

56, A121, M115, Y123 Linalool -- 2 A:B: I116 Y56, A121?? Which type of amino acid residues?

Write IC50 as IC50.

n=3, sample size too less.

Conclusion section replaced as Conclusion with future perspective.

Reviewer #2: The work is of sufficient importance and should be accepted for publication but the quality of the figures is not up to the mark, for this significantly improve the quality of all the figures. Moreover, the English language must be improved before accepted.

**Do you want your identity to be public for this peer review?** For information about this choice, including consent withdrawal, please see our Privacy Policy

Reviewer #1: No

Reviewer #2: **Yes: ** Muhammad Amjad Ali

---

## [Author Response · Author response to Decision Letter 1]

23 Apr 2025

Response Letter to the Reviewer Comments

We sincerely thank you and the reviewers for the detailed and constructive feedback on our manuscript. We have carefully addressed all comments and revised the manuscript accordingly. Below, we provide a detailed point-by-point response to each comment.

Reviewer 1 Comments:

We sincerely appreciate the reviewer’s feedback and have incorporated recent phytochemical and in silico related studies into the Introduction (Lines 75-91) (Lines 113-118), which has significantly improved the quality of the manuscript

Comment 1: Draw a figure for method section that’s why reader can easily understand.

Response: We appreciate the reviewer’s suggestion to include a visual representation of the methodology. To enhance clarity and reader understanding, we have now added a flowchart in the Methodology section that outlines the sequential steps of the study—from compound and target selection to in silico and in vitro analysis.

Comment 2: In results section figures, authors should add individual data points.

Response: Thank you for your valuable comment. Individual data points have now been added to Figures 5–6 and 15–16 to more accurately reflect experimental variability and enhance transparency in data presentation.

Comment 3: Authors should add novelty, and study limitation section before conclusion.

Response: Thank you for the valuable suggestion. We have now incorporated a brief section highlighting the novelty of our study, along with the key limitations, at the end of the Discussion section, just before the Conclusion.

Comment 4: Authors should be add a possible mechanism for this study.

Response: We appreciate the reviewer’s insightful comment. In the revised manuscript, we have included a discussion of the possible underlying mechanism, highlighting the signaling relationship between PTEN loss and increased PD-L1 expression. This supports the mechanistic basis of our findings and strengthens the overall impact of the study.

Comment 5: Authors should be add a section of docking validation.

Response: We appreciate the reviewer’s valuable suggestion. Docking validation has now been addressed within the molecular docking results section. We performed redocking and superimposed the predicted poses with crystallographic ligands to confirm the reliability of our docking protocol. The validation results, including key interacting residues, are presented in the supplementary material (Figure S1).

Comment 6: Mention Chemicals and reagents names.

Response: We sincerely appreciate your insightful suggestion. The names of all chemicals and reagents have now been listed in the Materials subsection (2.3.1).

Comment 7: Mention the concentration selection process.

Response: We are grateful for the reviewer’s thoughtful comment. The rationale for selecting concentrations ranging from 3.13 to 100 µg/mL was based on preliminary cytotoxicity screening and supported by relevant literature. This has now been clearly stated in the Treatment subsection (Section 2.3.4) and further discussed in the Discussion section (Lines 658–663).

Comment 8: Write the table and figure legends.

Response: Thank you for your valuable comment. The table and figure legends have been provided in accordance with PLOS ONE journal guidelines.

Comment 9: 56, A121, M115, Y123 Linalool -- 2 A:B: I116 Y56, A121?? Which type of amino acid residues?

Response: We express our gratitude for the reviewer's observation. In the revised manuscript, we have specified the residue types using three-letter codes or full names (e.g., Tyrosine (Tyr56)) to ensure consistency.

Comment 10: Write IC50 as IC50.

Response: We acknowledge the reviewer’s comment. This typographic issue has been corrected throughout the manuscript, and all instances are now consistently written as “IC₅₀”.

Comment 11: n=3, sample size too less.

Response: We appreciate the reviewer’s observation. However, we would like to clarify that n = 3 in our study does not refer to the sample size in the context of biological replicates or patient-derived samples. Rather, it denotes that the MTT assay was performed in technical triplicates, which is a standard practice to ensure the reproducibility and reliability of the assay results.

Comment 12: Conclusion section replaced as Conclusion with future perspective.

Response: We are grateful for the thoughtful insights. We have expanded the conclusion to discuss the potential future directions of this research.

Reviewer 2 Comments:

Comment 1: The work is of sufficient importance and should be accepted for publication but the quality of the figures is not up to the mark, for this significantly improve the quality of all the figures. Moreover, the English language must be improved before accepted.

Response: We sincerely appreciate the reviewer’s feedback. In response, we have significantly enhanced the quality of all figures using the PACE image optimization tool. Each figure has now been uploaded in high resolution (600 dpi) and in TIFF format, as per standard publication requirements. Additionally, the manuscript's English language has been thoroughly improved using the Paperpal editing tool to ensure clarity and readability

---

## [Editor Report · Decision Letter 1]

Linalool-Based Silver Nanoconjugates as Potential Therapeutics for Glioblastoma: In Silico and In Vitro Insights

PONE-D-25-10375R1

Dear Dr. Khan,

We’re pleased to inform you that your manuscript has been judged scientifically suitable for publication and will be formally accepted for publication once it meets all outstanding technical requirements.

Kind regards,

Mansureh Ghavam

Academic Editor

PLOS ONE
---

## [Editor Report · Acceptance letter]

PONE-D-25-10375R1

PLOS ONE

Dear Dr. Khan,

I'm pleased to inform you that your manuscript has been deemed suitable for publication in PLOS ONE. Congratulations! Your manuscript is now being handed over to our production team.

Kind regards,

on behalf of

Dr. Mansureh Ghavam

Academic Editor

PLOS ONE